# Gastroesophageal Reflux Disease and Foregut Dysmotility in Children with Intestinal Failure

**DOI:** 10.3390/nu12113536

**Published:** 2020-11-18

**Authors:** Anna Rybak, Aruna Sethuraman, Kornilia Nikaki, Jutta Koeglmeier, Keith Lindley, Osvaldo Borrelli

**Affiliations:** 1Department of Gastroenterology, the Great Ormond Street Hospital, Great Ormond Street, London WC1N 3JH, UK; aruna.sethuraman@gosh.nhs.uk (A.S.); jutta.koeglmeier@gosh.nhs.uk (J.K.); keith.lindley@gosh.nhs.uk (K.L.); osvaldo.borrelli@gosh.nhs.uk (O.B.); 2Wingate Institute of Neurogastroenterology, Blizard Institute, Barts and The London School of Medicine and Dentistry, QMUL, 26 Ashfield Street, Whitechapel, London E1 2AJ, UK; kornilia.nikaki@nhs.net

**Keywords:** GERD, intestinal failure, short bowel syndrome, dysmotility, pediatric intestinal pseudo-obstruction syndrome, gastroparesis, nutrition, children

## Abstract

Gastrointestinal dysmotility is a common problem in a subgroup of children with intestinal failure (IF), including short bowel syndrome (SBS) and pediatric intestinal pseudo-obstruction (PIPO). It contributes significantly to the increased morbidity and decreased quality of life in this patient population. Impaired gastrointestinal (GI) motility in IF arises from either loss of GI function due to the primary disorder (e.g., neuropathic or myopathic disorder in the PIPO syndrome) and/or a critical reduction in gut mass. Abnormalities of the anatomy, enteric hormone secretion and neural supply in IF can result in rapid transit, ineffective antegrade peristalsis, delayed gastric emptying or gastroesophageal reflux. Understanding the underlying pathophysiologic mechanism(s) of the enteric dysmotility in IF helps us to plan an appropriate diagnostic workup and apply individually tailored nutritional and pharmacological management, which might ultimately lead to an overall improvement in the quality of life and increase in enteral tolerance. In this review, we have focused on the pathogenesis of GI dysmotility in children with IF, as well as the management and treatment options.

## 1. Introduction

Modern parenteral nutrition (PN) and surgical and medical management of intestinal failure (IF) have resulted in a significant increase in the long-term survival of children with IF in recent decades [1]. Morbidity related to impaired motor activity of the upper gastrointestinal (GI) tract is high in children with congenital or acquired short bowel syndrome (SBS) and those with disturbance of the function of the enteric nervous system or muscular layer of the gut. According to the literature, up to 40–50% of patients with gastroschisis and 79–90% with intestinal malrotation [2] have gastroesophageal reflux disease (GERD) and esophageal dysmotility [2,3,4]. In pediatric intestinal pseudo-obstruction (PIPO) syndrome, the small intestine is always involved, but the esophagus, stomach and colon may also be affected [5]. Moreover, many of the IF causes overlap; for example, 30% of patients with PIPO have malrotation [6]. Intestinal atresia and gastroschisis are also reported associated comorbidities in PIPO [7]. In addition, prematurity, chronic lung disease and congenital heart disease are also seen in children with IF, particularly in those with SBS.

Dysmotility of the GI tract in IF can be a consequence of a primary GI neuromuscular disorder, such as in PIPO, alterations in GI anatomy, defects in enteric hormone secretion and GI tract denervation, which may result in rapid intestinal transit, ineffective antegrade peristalsis, delayed gastric emptying or gastroesophageal reflux.

Normal gastrointestinal motility is dependent upon the coordinated function of three elements namely interstitial cells of Cajal (ICC), the enteric nervous system (ENS) and intestinal smooth muscle. ICCs have a number of different functions including electrical pacemaker activity generating electrical slow wave activity and transducing motor neural inputs from the ENS to the GI musculature [8]. ICC are distributed throughout the whole GI tract—from the distal esophagus to the anal sphincter. Changes in ICC distribution, morphology and density have been described in several conditions associated with IF including patients with gastroschisis, intestinal atresia (in the segment proximal to the atretic part), PIPO and Hirschsprung’s disease [8,9,10,11]. A summary of the impact of particular conditions, related to IF, on GI motility is presented in Table 1.

The mechanisms leading to GI dysmotility are complex. Understanding the pathophysiology of the disorders and its impact on the GI tract, helps to guide clinicians and prioritize appropriate selection of investigations available to the units. It also tailors nutritional and medical management according to the individual patient’s needs, without unnecessary use of diagnostic tools or treatments, leading to overall improvement in quality of life of this heterogeneous group of patients.

This review focuses on the foregut dysmotility manifestations, pathophysiology and treatment in children with IF, with the emphasis on the gastroesophageal reflux (GERD).

Using the Medline database, the keywords related to IF, dysmotility, GERD and pediatric population were used. We also searched and reviewed the references of the selected published articles.

## 2. Foregut Manifestations and Their Pathophysiology in Intestinal Failure

IF is a complex disorder, with diverse underlying causes, as described in the other chapters of this supplement. Foregut dysmotility and feeding difficulties are related to the underlying cause of IF, comorbidities as well as complications of the treatment itself (pharmacological, nutritional and surgical).

### 2.1. Oral Aversion and Dysphagia

In clinical practice, SBS is divided into three phases: an acute phase, an adaptation and a recovery phase [17]. Successful bowel adaptation may lead to restoration of enteral autonomy. Intestinal adaptation is improved with early oral and enteral feeding [18]. Oral feeding stimulates salivary glands which activate the secretion of an epidermal growth factor (EGF) and other trophic factors. It should be introduced as early as possible to stimulate oral motor development and to avoid oral aversion [19]. Children with SBS may display oro-motor, sensory and developmental feeding problems and resistance to oral feeding. Three main issues are identified causing oral aversion in children with SBS namely the physical, developmental and social aspects of eating and mealtimes [18].

The physical aspect of oral feeding includes lack of positive oral feeding experiences and changes in hunger satiety patterns. Many children with SBS/IF are at risk of developing oral aversion due to exposure to oral aversive stimuli; for example, nasogastric tubes, prolonged airway management, airway and upper gastrointestinal tract suctioning [20]. Children with SBS receiving most of their nutrition via enteral feeding tubes or as PN will not experience normal hunger-satiety patterns [21]. Enteral feeding, especially when provided over 24 h, and PN interrupt hunger-satiety patterns that are important learning experiences, although cyclical PN is advocated as early as possible mainly to avoid long term PN complications [22].

Developmental delay in feeding, oral motor and sensory processing difficulties are seen in children with SBS mainly due to decreased feeding experience and loss of critical windows of opportunity for establishing normal suck and swallow patterns. Initiation of solid food diet is an important stage in development and nutrition and deprivation of these experiences delay acquisition of optimal feeding skills [23,24]. Moreover, social factors, such as reduced participation in family mealtimes, might contribute to delay in oral feeding. Participating in regular family meals can be challenging for caregivers of children with SBS because of time taken to look after a child with high medical needs and repeated episodes of hospitalization. These may disrupt the bonding and attachment process and contribute to parental stress. Reduced participation at mealtimes also reduces exposure to the typical cultural and social aspects of mealtimes not only with the family but also with their peers [25].

Gastroesophageal reflux (GER) disease (GERD) has a high prevalence in IF. In a cohort of 700 children described by Rommel et al., with diverse medical conditions, referred for assessment of severe feeding difficulty, it was the most frequent organic cause of feeding problems, affecting 33% of children in this study [26]. The importance of GERD as a causal factor of oral aversion and feeding difficulties is also underlined by a recent study examining feeding difficulties as a core outcome for GERD [27].

Dysphagia in infancy (abnormalities of swallowing), represents a major problem in children with gastroschisis and in other groups of high-risk infants (premature birth, low-birth-weight, congenital anomalies, perinatal asphyxia, postsurgical, and sepsis) [28]. Jadcherla et al. compared a small group of infants with gastroschisis with normal controls demonstrating that basal pharyngoesophageal peristaltic failure was among the most common detected mechanisms of esophageal dysmotility [29]. These patients needed longer enteral nutrition support.

Lesions in the oral cavity, pharynx and larynx due to prolonged endotracheal intubation can impair the oro-pharyngeal motility and sensitivity and impact on the swallowing process, thus resulting in oropharyngeal dysphagia [30].

### 2.2. Regurgitation and GER in Children

GER is the result of an imbalance between the noxious stimuli produced in the esophagus, such as reflux events and refluxate characteristics, versus its defense mechanisms, such as esophageal clearance and mucosal sensitivity [31]. In the context of IF, some of these factors are altered, resulting in the symptoms of reflux disease—Table 2.

There are limited data regarding the changes on pH-metry and pH-impedance in children with IF. Dermibilek et al. reported that esophageal pH monitoring studies demonstrate a significantly higher acid exposure time (AET) in children with concomitant GERD and intestinal malrotation (mean AET: 14.7%) versus those with isolated GERD (mean AET: 7.06%) [32].

Erosive esophagitis in children with gastroschisis is documented in 15% of cases [33], compared with a reported incidence for the general 0–5 years pediatric population (5.5%) [34]. Data in both SBS and IF are scanty, possibly due to early introduction of treatment with anti-acid drugs, limiting the availability of data on GERD investigations and affecting the epidemiology. In a highly selective group of 56 children (median age 5.8 years) presenting to an intestinal rehabilitation center, esophagitis was reported in 15% [35], with PIPO having a higher incidence (37.5%).

The function of the esophageal epithelial barrier in IF has not been evaluated, but in general, it is expected to be compromised where there is inflammation and ischemia. Similarly, the role of peripheral and central hypersensitivity in children with SBS and IF has not been studied.

#### 2.2.1. Impact of the Distorted Anatomy on GERD

Transient lower esophageal sphincter relaxations (TLESRs) occur more frequently when there is an increase in intragastric pressure, which can arise in children with SBS either due to surgical reasons or as an effect of delayed gastric emptying. It is well known that GER episodes occur more often in the presence of increased number of TLESRs [36].

The incidence of hiatus hernia in patients with complex gastroschisis has been reported as 18.9% [37], which is similar to the general pediatric population referred for upper endoscopy [38]. In the same cohort of patients, 37.9% of cases with complex gastroschisis underwent fundoplication for severe GERD, half of them also had a hernia repair [38]. The angle of His can also be distorted from operative reduction of gastroschisis [39]. Distortion of the oblique gastric sling fibers forming the angle of His is associated with severe GERD [40].

In children with gastroschisis, a number of esophageal motor abnormalities have been described compared to controls: a. lower frequency and poor propagation of spontaneous swallows; b. slower esophageal peristaltic propagation velocity; c. impairment of reflexes involved in primary peristalsis, secondary peristalsis, upper esophageal sphincter contractility and lower esophageal relaxation [29]. These abnormalities likely lead to impaired reflux clearance and increased occurrence of feeding difficulties and respiratory complications. The cause of these abnormalities may lie in the central nervous system as well as in the enteric nervous system. In a rat model of gastroschisis, the enteric nervous system displayed delay in neuronal differentiation and myenteric plexus organization [41].

In children with megacystis-microcolon-intestinal hypoperistalsis syndrome, the lower esophageal sphincter has a normal resting tone and relaxation but there is absent peristalsis in the esophageal body with or without concomitant esophageal dilatation [42].

Up to 25% of patients investigated for GOR may have concomitant malrotation [32,43]. Gastric emptying is significantly prolonged in children with GERD and intestinal malrotation [32]. This may be secondary to partial gastric outlet and duodenal obstruction, although the data are conflicting as to whether a Ladd’s procedure suffices for the resolution of GORD symptoms, without the need of fundoplication [44,45].

#### 2.2.2. Complications Related to Necrotizing Enterocolitis

In necrotizing enterocolitis (NEC) the combination of ischemia, inflammation, bacterial overgrowth and reparative tissue changes lead to multi-layer involvement [46]. In turn, these reparative changes, which include vascular degeneration and intestinal fibrosis, lead to segmental dilatation and/or stenosis and suppression of migrating motor complex activity. Therefore, in SBS following NEC, the remaining bowel may display both impaired absorptive capacity and dysmotility. In experimental mice, gastric emptying delay is correlated to the severity of intestinal damage [47].

#### 2.2.3. Impact of Small Bowel Resection on Gastric Acid Secretion

Extensive enterectomy is associated with a transient hypergastrinemia and gastric hypersecretion [48]. Gastric acid secretion has a cephalic phase and a gastric phase. In the cephalic phase the acid secretion is triggered by acetylcholine released by the vagal afferent nerves, while in the gastric phase the trigger is induced by the entry of nutrients in the GI tract and it is mediated by gastrin produced by antral G cells [49]. In SBS, gastric acid hypersecretion impairs intraluminal digestion. The excessive acid that enters the proximal small bowel inactivates pancreatic enzymes, impairs micelle formation and lipolysis, and reduces polymeric carbohydrates digestion, resulting in malabsorption [50]. Treatment of gastric acid hypersecretion with H2 receptor antagonists and/or proton pump inhibitors has improved nutrient absorption with SBS.

Gastric acid hypersecretion is also a well-known cause of esophageal and peptic ulcers in SBS [48]. Gastric acid hypersecretion seems to persist till 6–12 months, postoperatively, and it is caused by lack of inhibitory hormones produced in the proximal gut (i.e., gastric inhibitory peptide and vasoactive intestinal peptide). Moreover, gastric acid hypersecretion is likely to have also an effect on the pepsin precursors leading to pepsin activation [51]. Activation of pepsin in the refluxate will lead to more significant esophageal mucosa impairment and reflux disease.

Basal gastric acid hypersecretion was found to be associated with two factors: extensive small bowel resection and also initiation of enteral feeding [50].

Bile acid deconjugation in SBS occurs as a result of *Lactobacilli* activity [52], leading to a disturbed enterohepatic circulation [53] and possibly to an alteration in the bile acid composition of the gastric juice. Both conjugated and unconjugated bile acids cause mucosal impairment of the esophageal mucosa. Either way, duodeno-gastric reflux due to dysmotility of the proximal duodenum [54] increases the content of bile acids in the gastric refluxate to the esophagus. The latter is a well described risk factor for erosive esophagitis and Barrett’s esophagus [55].

### 2.3. Delayed Gastric Emptying

There is paucity of pediatric data on prevalence of gastroparesis or delayed gastric emptying in IF patients. In two large pediatric series, gastroparesis was associated with surgical procedures in 12.5% of children with delayed gastric emptying [56,57]. Among the IF cohort, surgical procedures are due to bowel anatomic irregularities such as malrotation [32], NEC, gastroschisis, intestinal atresia, less commonly fundoplication and gastrostomy placement [58]. Motility disorders are also associated with delayed gastric emptying.

Normal gastric emptying is a highly complex and coordinated process, which includes proximal stomach accommodation, antral contractions, pyloric sphincter relaxation and antro-pyloric-duodenal coordination [59]. Gastroparesis may occur due to disturbance of the above mechanisms, including altered fundic receptive relaxation, decreased antral contractility and incoordination of gastric emptying and duodenal contractions. Slow fundic contractions help in the transfer of gastric contents from the fundus to the antrum. These contractions might be affected by gastrostomy placement in the gastric body [60] and, in pseudo-obstruction syndrome, by depleted ICC cells affecting the electric activity of the smooth muscle cells in antrum [61].

Nausea, vomiting, abdominal pain, bloating, early satiety and weight loss are some of the common symptoms of gastroparesis. Infants and young children commonly present with vomiting, whereas adolescents present frequently with nausea and abdominal pain [56]. No single test studies all aspects of gastric motility. Poor standardization of diagnostic tests and paucity of pediatric normative data makes diagnosing gastroparesis in children a challenge. The current gold standard in the diagnosis of gastroparesis is determined by the demonstration of delayed gastric emptying with a solid meal with 4-h gastric scintigraphy. Gastric emptying breath test is also available, which uses the nonradioactive isotope ^13^C bound to octanoic acid to label the solid component of a test meal [62]. Wireless motility capsule and electrogastrography are also used; however, their role is still confined to the research area.

Understanding the mechanisms of delayed gastric emptying in children with IF contributes to the medical management as well as route of enteral feeds.

### 2.4. Impact of Small Bowel Resection on Motility

SBS is a form of IF resulting from surgical resection, congenital defects or diseases with loss of absorptive surface area [63]. The most common etiologies of SBS are NEC (45%), gastroschisis (23.8%), intestinal atresia (17.5%) and midgut volvulus (17.5%) [64]. In this cohort of children with SBS, the neuromuscular dysfunction of the GI tract may involve the whole bowel or may be segmental. It has been suggested that both ischemic damage to the enteric nervous system and injury to the smooth muscle cells may contribute to motility abnormalities [12]. Dysmotility following surgical resection is reported in 43% of children with gastroschisis, 50% of children with intestinal atresia (50%) and between 8% and 12% of children with NEC [65,66]. The diagnosis of dysmotility in these subsets of children has been primarily based on symptoms like vomiting, abdominal pain, feeding intolerance, recurrent episodes of abdominal distension, diarrhea and bacterial overgrowth [67]. Antro-duodenal and colonic manometry in SBS are not being performed regularly, due to limited availability. Upper GI contrast studies are frequently performed to assess for evidence of strictures and intestinal dilatation and to assess transit times.

Intestinal atresia may be observed in isolation or may be found in 10% to 20% of patients with complex gastroschisis and may be associated with a more severe form of dysmotility [65]. Dysmotility is a particularly vexing problem with small intestinal atresia. The etiology is associated with a segmental absence of muscular layers with fibrous replacement, alternating with segmental hypertrophy of the muscle layers in the proximal atretic segment [68]. Corresponding hypoplasia, immaturity, and/or absence of ganglia in the myenteric plexus are observed on both the mesenteric and antimesenteric sides in the same region as the defective muscularis. Masumoto et al. demonstrated alterations in the myenteric nerves, ICC and smooth muscle cells, and they suggested that these changes are secondary to an acute ischemic event [68]. This segmental pattern leads to localized dilation with obstruction caused by interruption of the migrating motor complex (MMC) and, therefore, abnormal peristalsis.

The pathologic changes of NEC suggest that it is a form of ischemic bowel disease. NEC associated with massive intestinal resections will result in rapid intestinal transit. The etiology of this mechanism, as showed by Husebye et al., is complex and includes: loss of the ileal braking mechanism, perturbations of the MMC with a loss of the normal balance between absorption and intestinal clearance and segmental areas of dysmotility arising from ischemia and stricture formation complicated by the bacterial overgrowth [69]. Experimental studies on animal models have shown that distal resections of the small bowel, typical in NEC, exhibit more severe dysmotility and clinical problems when compared with proximal resections of comparable length [70]. Spiller et al. explain this phenomenon by a loss of the ileal inhibitory control, which contributes to accelerated transit through the proximal bowel [71]. Studies suggest that the histopathologic alterations within the intestinal wall associated with delayed or altered maturation of the ICC could explain the dysmotility observed in gastroschisis. Understanding the histopathological changes and basis of dysmotility helps in further medical and surgical management.

## 3. Impact of Feeding on GI Motility in IF

Optimal nutrition is very important in children with IF for a number of reasons: a. intraluminal nutrients have stimulatory effects on epithelial cells and on trophic hormones that enhance intestinal adaptation [72]; b. introduction of PN greatly improved the prognosis and long-term outcomes [73]; c. good nutritional status of patients, as well as adjusting the macronutrients in enteral feeding, promote gastrointestinal motility [74].

An oral diet, if safe for the patient, should be encouraged, as it is part of normal neuropsychological development and can prevent oral aversion and feeding difficulties [74]. Based on the patient’s skills, bowel absorption surface and extent of GI dysmotility, oral or enteral diet can be based on either or combination of taste stimulation only, foods that dissolve in the mouth before swallowing, liquid diet or a diet with an adjusted macronutrient content for promoting motor activity of the GI tract. Whilst early oral or enteral feeding is promoted, the optimal route (oral bolus versus enteral bolus or continuous feed), as well as the type of feed to promote optimal adaptation, with the ultimate goal of enteral autonomy and weaning from PN, is currently not fully understood, and further research is needed.

Gastric emptying is influenced by a number of factors, including meal volume, caloric content, fat and fiber content, as well as meal composition (solid vs. liquid) [75]. Both fat and fiber tend to delay gastric emptying [76]. A study by Van Den Driessche, comparing gastric emptying (assessed with ^13^C-octanoic acid breath test) in infants with breast milk versus polymeric formula, indicated faster gastric emptying of human milk [77], which was potentially linked to breast milk being a whey dominant feed with low osmolality [78]. Emptying of a liquid feed is more rapid than a solid meal as the physiology of liquid and solid emptying is quite different [79]. In adults, stomach emptying rate was reported to be approximately 1–2 kcal/min; therefore, increasing the liquid nutrient component of meals and keeping small meal size should be advocated in patients with delayed gastric emptying [80].

There is a paucity of evidence-based data on the type of enteral formula feeding on the gastro-intestinal motility. In the most severe presentation of dysmotility in IF, such as PIPO, different strategies, such as oral feeding, enteral feeding (bolus or continuous) or PN, should be tailored to each patient, depending on extent of motility disturbance and feeding tolerance [6]. Depending on the degree of dysmotility, continuous feed (gastric or jejunal) might be preferred option, as well as the use of hypoosmotic and hydrolyzed formulae [6]. The group of Di Lorenzo showed that presence of migrating motor complex (MMC) on antro-duodenal manometry is a predictor of good tolerance of jejunal feeding in patients with PIPO [81].

With regards to the enteric adaptation, the use of human milk or amino-acid formulas seem to be best tolerated and is associated with shorter duration of PN [82].

Despite paucity of data in the pediatric group, there is increasing interest in the blended diet in patients with GI dysmotility. Single studies showed a positive impact of blenderized diet on the diversity and richness of gut microbiome [83], which could potentially lead to a positive impact on feeding tolerance in severe gut dysmotility [84]. A blenderized diet can improve enteral feed volume tolerance, reducing upper GI symptoms, such as retching, gagging and vomiting [85]. Samela et al. showed improved stool output in children with IF, when transitioned from elemental formula to the tube feeds with real food ingredients [86]. More research is needed to assess the composition and safety of blended diet in children with IF associated dysmotility.

It is worth noting that it is not only the length of the remaining small bowel, but also the presence and continuity with the colon, particularly the right colon, impacts the adaptation process and the prospect of enteral autonomy. Lambe et al. demonstrated that in the presence of colon, the absorption rate increases two fold, for the same small bowel length [87]. This is likely due to a number of factors including regulation of motility, regulation of small bowel adaptation and colonic salvage.

Nevertheless, a large number of IF patients will require total or partial parental nutrition (PN) to optimize nutritional status. Approximately two thirds of PIPO patients are PN-dependent [6]. The success rate of PN weaning is higher among children with SBS (42–73%) [88]. In a retrospective 17 year follow-up of neonatal onset SBS, 76% of patients achieved intestinal autonomy [89].

## 4. Tools in the Diagnostics of the Upper GI Dysmotility

### 4.1. Barium Contrast Study

While pH-impedance is considered as the gold standard for the diagnosis of GORD, an upper GI contrast study has a sensitivity of 42.8% and a negative predictive value of 24% [90]. Therefore, a contrast study has a place to detect structural abnormalities but cannot be used in isolation to diagnose GORD [91]. In the context of IF, an upper GI contrast study has a role in evaluating the presence or absence of hiatus hernia, duodenal dilatation or stenosis and malrotation.

### 4.2. Esophago-Gastro-Duodenoscopy

When upper GI endoscopy is employed in patients with GER symptoms, the endoscopic findings that are pathognomonic for GERD include erosions, strictures and Barrett’s esophagus [92]. Endoscopy has an excellent specificity, but its value is hampered by its poor sensitivity as the majority of cases present with non-erosive reflux disease [93]. On the other hand, it has been shown that endoscopy is particularly useful in children with IF [35,94].

### 4.3. 24-Hour Esophageal pH-Impedance Study

Although there is no “gold” standard diagnostic facility for GERD, pH-metry and/or pH-impedance studies are regarded as pivotal in the diagnostic algorithm of GORD [95]. pH-impedance is gaining ground and is now the investigation of choice in many specialist centers as it facilitates the GERD phenotyping according to the Lyon consensus criteria [96,97].

### 4.4. High Resolution Esophageal Manometry

High resolution esophageal manometry in the remits of GERD is undertaken as a routine procedure in adult practice in order to: a. locate the lower esophageal sphincter (LES) and facilitate the correct placement of the reflux monitoring catheter; b. exclude the presence of achalasia; c. assess the esophageal motility prior anti-reflux surgery; d. differentiate rumination syndrome for GERD [95]. Moreover, it provides supportive evidence for GERD when hypotensive LES, hiatus hernia or esophageal hypomotility are detected [96]. In pediatrics, esophageal manometry has similar indications [98], but is applied more selectively as it is considered invasive. Recently, the oro-pharyngeal impedance manometry has become accessible for the assessment of the swallowing physiology [99].

### 4.5. Antro-Duodenal Manometry

Antro-duodenal manometry (ADM) is the investigation of choice for the diagnosis of foregut dysmotility and aims to describe whether the dysmotility is due to a myopathy, neuropathy or unproven mechanical obstruction [98,100]. It assesses the antral and small bowel neuromuscular function by measuring the contractile activity during the fasting period and in response to either test meal or stimulant medicines. Interpretation of ADM is challenging for a number of reasons: a. pediatric normative data are not available; b. atypical patterns can present in health; c. age influences the manometric measurements; d. eliciting a manometric response in the post-prandial period is dependent on adequate caloric intake, which may be impossible due to patient’s symptoms; e. artefacts from movement need to be recognized and excluded from analysis [98]. ADM helps in predicting the outcome in pseudo-obstruction [101,102], the response to prokinetics [103] and the need for intestinal transplantation [104].

### 4.6. Nuclear Medicine Gastric Emptying Study

Gastric emptying scintigraphy, performed with a standardized meal and study protocol [105], is the most commonly used test to directly assess gastric motility [106]. However, it involves radiation and there is a 12–15% intraindividual variability [107]. Most importantly, the correlation between abnormal findings and patient symptoms is not well established [108]. Stable isotope breath tests are an alternative to gastric emptying scintigraphy, but they appear to be less accurate in the presence of reduced absorptive capacity of the intestinal mucosa, and in the pancreatic, liver or respiratory disorders and, most importantly, they lack standardization of analysis [106]. Magnetic resonance imaging of the stomach, functional ultrasonography, tests for gastric capacity and accommodation, single photon emission computed tomography (SPECT), satiation or nutrient drink test and gastric myoelectrical activity are not widely used either due to the highly specialized equipment and skills required or the lack of normative data and clinical applicability of abnormal findings [106]. Wireless pH and motility capsules [109] are not generally employed in the pediatric population in IF in view of the possibility of capsule retention due to strictures or hypomotility.

Table 3 presents diagnostic tools in foregut dysmotility with the most common findings in the IF/SBS.

## 5. Management of GERD and Foregut Dysmotility in IF

### 5.1. Non-Pharmacological and Dietary Treatment

A multidisciplinary approach is important in the evaluation and management of children with IF, complex feeding difficulties and GI motor disorders. Most teams include physicians, nurses, dieticians, occupational therapists, psychologists, physical therapist and social workers [110]. An early approach with the use of non-nutritive oral motor exercises and sensorimotor interventions to improve alertness, suck strength and organization, will help to facilitate transition from enteral to oral feeding in the future [111].

With regards to foregut dysmotility, a conservative approach is a first line treatment. Modifications of the feeding frequency and volume, child’s position during meals and changes of the diet (e.g., 2–4 weeks trial of dairy free diet, use of thickeners) will prove sufficient in some patients. Complementary treatments such as prebiotics, probiotics, or herbal medications to treat GERD are not currently recommended [112]. Other nutritional management strategies are described in the paragraph *Impact of feeding on GI motility in IF*. These include optimizing meal volume, caloric content, fat and fiber content, as well as meal composition (solid vs liquid) to promote stomach emptying and small bowel transit. In case of alarm signs, such as bilious vomiting, GI bleeding, failure to thrive and lethargy, between the others, an appropriate investigation of symptoms is necessary [82,112,113].

### 5.2. Pharmacological Treatment

Pharmacological treatment in pediatric GERD and foregut dysmotility is based on the use of proton pump inhibitors (PPI), histamine-2 receptor antagonists (H2 blockers) and prokinetic drugs. The use of these agents, either single or in combination, should be reserved for patients with objectively assessed symptoms.

As mentioned above, extensive resection of the bowel is associated with chronic acid hypersecretion and hypergastrinemia. Duration of hypersecretion may be related to the extent of bowel resection and is transient. Other factors, such as absence of inhibitory substances released from small intestine, may also mediate hypersecretion [50]. Treatment of gastric acid hypersecretion with H2 blockers and/or PPIs has improved nutrient absorption in SBS. H2 blockers were also proved to be superior to placebo in healing erosive esophagitis in pediatric populations [114]. The optimal duration of treatment is speculative and is not clear [115]. Data suggest a link between acid blockade and increase in respiratory and gastrointestinal infections [116], including bacterial overgrowth. Therefore, it is worthwhile to wean patients from acid blockade after some time. Moreover, unlike PPIs, H2 blockers exhibit tachyphylaxis and tolerance, therefore they should not be considered in the long-term management of GERD. PPIs are more effective in treating erosive esophagitis in comparison with H2 blockers, but their overprescribing is a concerning issue, particularly in the youngest group of patients. Detailed guidelines on clinical management of GERD in infants and older children are described in the joint recommendations of the North American Society for Pediatric Gastroenterology, Hepatology and Nutrition (NASPGHAN) and the European Society for Pediatric Gastroenterology, Hepatology and Nutrition (ESPGHAN) [117].

In a systematic review by Dicken et al., authors collected data on the presence and distribution of the gastrointestinal receptors in the GI tract, including dopamine, serotonin, motilin and opioid, giving the basis of prokinetic use in foregut motility disorders [67] (Figure 1). The most commonly used prokinetic agents, with their activity site and evidence, are presented in Table 4, but overall, there is a paucity of data supporting promotility agents, especially in pediatric patients with IF or SBS. Despite that, medicines, such as domperidone or metoclopramide are used in children with GERD, feeding difficulties and delayed gastric emptying [118]. There is no unified guideline on the duration of the treatment with prokinetic drugs and there is a great variety among the published data [118]. In the systematic review of use of domperidone in GERD in children, the treatment duration varied between 2 and 8 weeks [119]. Trial with prokinetic treatment should be followed by clinical monitoring, to avoid unnecessary use of ineffective treatment. Monitoring should include symptoms control (e.g., reduction in regurgitation frequency, decrease in gastric aspirates), as well as ability to increase enteral feeding rate or oral intake. The use of prokinetics should balance potential clinical benefits with serious side effects, including dystonic reactions and seizures (metoclopramide), cardiac risk and hyperprolactinemia (domperidone), as well as potential interactions with other medications.

### 5.3. Indication for Surgical Management

There are certain conditions and motor disturbances in IF, in which surgery will play a pivotal role.

With regards to antireflux surgery, it is reserved for patients with intractable or relapsing reflux, and for patient with high risk of the GERD complications, including aspiration pneumonia or exacerbation of asthma [117]. On the other hand, one need to balance the fact that children who have conditions predisposing to severe GERD are the same group of children who have the highest risk for postoperative complications. Patients with respiratory complications of GERD appear to benefit most from surgical treatment [140]. In a 20-year experience review of laparoscopic Nissen’s fundoplication in children, Rothenberg et al. showed that the highest rate of the fundoplication wrap failure was in the group of infants below 6 months of life [141]. The most common, although rare, complications were prolonged gastroparesis and significant dysphagia. The recurrence rate and the need for a redo procedure is higher in patients younger than a year of age. Furthermore, a higher rate of complications is observed in a redo of the anti-reflux surgery [141]. Fundoplication, per se, not infrequently can cause or worsen esophageal motor function.

Classical surgical techniques in GI dysmotility encompass the provision of access to the stomach and small bowel in the form of a gastrostomy or enterostomy (i.e., jejunostomy or ileostomy), to support enteral feeding or access for upper GI depressurization and venting [6,142]. Certain gastrostomy devices can be now easily converted to gastro-jejunostomy appliance and, therefore, give access for direct small bowel feeding.

In delayed gastric emptying, with preserved small bowel motor function, implantation of the gastric pacemaker, with continuous high frequency and low energy electrical stimulation, was shown to be effective for symptoms such as vomiting and nausea in adult patients [143]. To date, data on the gastric pacing use in children are scarce. In one study on nine patients with a mean age of 14 years, suffering from chronic nausea and vomiting, with delayed gastric emptying for solid meal, electrical stimulation of the stomach significantly alleviated symptoms, but did not change gastric emptying parameters [144]. The group of Carlo Di Lorenzo showed similar results, noting no serious adverse events [145].

Intrapyloric injection of the Botulinum toxin A has also been described in children with gastroparesis. In an open label study of 45 children, the response rate after first injection was 66.7% and the median duration of response of 3 months [146]. Within this group authors describe only one child with side effect of treatment (exacerbation of vomiting).

In SBS, the timely reestablishment of the bowel continuity and closure of the stomas support obtaining enteral autonomy and acquisition of the enteral feeding tolerance [147,148]. In PIPO creation of a defunctioning ostomy can provide adequate gastrointestinal decompression and increase enteral feeding tolerance [6].

## 6. Summary and Conclusions

The pathophysiology of the disturbed foregut motility in IF is complex and varied depending on many factors, including underlying condition, extent of the bowel resection and impaired function and communication between gut pacemaker cells, ENS and smooth muscles of the bowel. Foregut dysmotility and feeding difficulties related to it are also related to the complications of the treatment IF/SBS itself, including pharmacology or surgical procedures. The complexity is best presented based on the effects of IF on the development and severity of GERD. The hypersecretion of the gastric acid, distorted anatomy affecting the angle of His, abnormal motility of the stomach and esophagus are only some of the symptom triggers and modulators.

In severe conditions, such as PIPO, treatment can be very challenging, and many patients will remain dependent on PN to control their symptoms and support optimal growth. Decisions on the management should be guided by gold standard diagnostics and multidisciplinary discussion and should be tailored to the patient’s needs.

## Figures and Tables

**Figure 1 nutrients-12-03536-f001:**
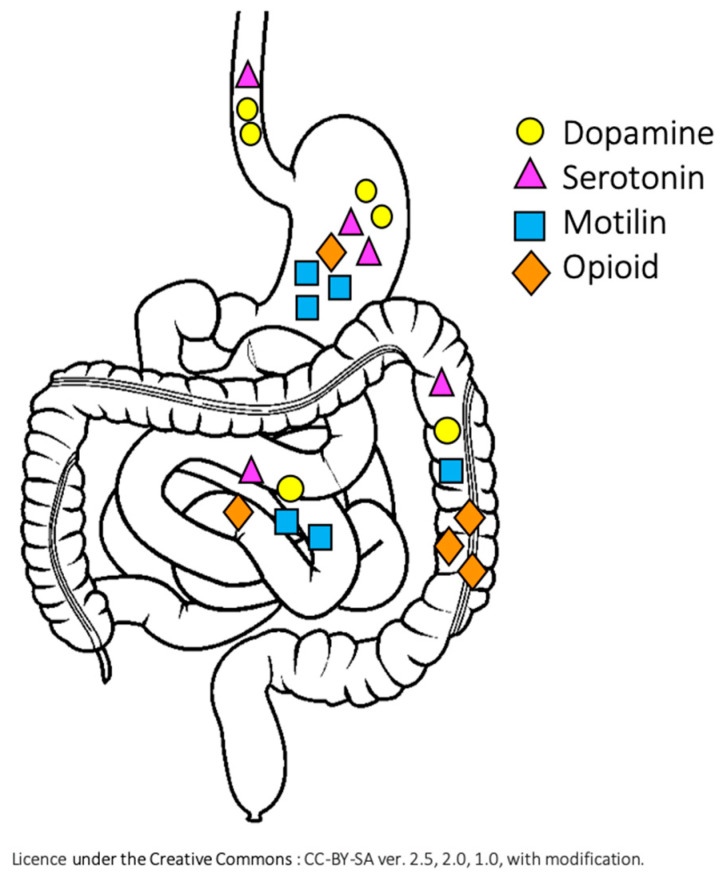
Distribution of the motility-related receptors in the gastrointestinal tract [67].

**Table 1 nutrients-12-03536-t001:** Causes of intestinal failure (IF) in children, which impact on gastro-intestinal motility.

Causes of IF	Possible Mechanisms Contributing to GI Dysmotility
**Necrotizing enterocolitis**	Gastric hypergastrinemia and hypersecretion in extensive gut resection. Ischemic injury to the enteric nervous system or damage to the smooth muscle cells.
**Gastroschisis**	Oro-pharyngeal dysphagia. GERD. Hiatal hernia. Severe intestinal dysmotility in >30% of patients. Decreased ICC [11]. Damage of the smooth muscle cells due to bowel wall ischemia [12].
**Malrotation and midgut volvulus**	GERD and gastric dysmotility. Small bowel dysmotility with neuropathic changes [13].
**Intestinal atresia**	Severe dilatation of the proximal bowel with hypoperistalsis, reduction of the intramuscular nerve fibers [14]. Decreased ICC [15].
**Intestinal aganglionosis**	Dysfunction of the ENS. Absent or sparse ICC in aganglionic and ganglionic bowel [16].
**PIPO**	Small bowel always affected. Potential involvement of esophagus, stomach and colon, with dysphagia, gastroparesis and constipation. More severe disease course in myopathic type of PIPO. Decreased or absent ICC.

IF—intestinal failure; GI—gastrointestinal; ICC—interstitial cells of Cajal; PIPO—Pediatric intestinal pseudo-obstruction syndrome; ENS—enteric nervous system; GERD—gastro-esophageal reflux disease.

**Table 2 nutrients-12-03536-t002:** Gastroesophageal reflux symptom triggers and modulators in intestinal failure.

Symptom Triggers		Effect of IF
Gastroesophageal reflux events	Number of events	Increased-Impaired gastric emptying
	Gas/liquid composition	Liquid-Impaired gastric emptying
	Volume refluxed	Larger—more proximal events with longer clearance time Impaired gastric emptying
Constituents of gastric juice	Acid	Gastric acid hypersecretion and hypergastrinemia
	Bile acids	Duodeno-gastric reflux of bile acids—abnormal concentration/composition of bile acids
	Pepsin	Activated
**Symptom Modulators**		
Refluxate clearance	Hiatus hernia	Similar to general population
	Hypotensive sphincter	Distortion of anatomy/Altered anatomy of the angle of His Increased frequency of TLESRs
	Peristaltic vigor	Slow/Absent propagation of esophageal peristalsis
Tissue sensitivity	Epithelial injury	Higher incidence of GERD in gastroschisis
	Central hypersensitivity	No data available for IF
	Peripheral hypersensitivity	No data available for IF

IF—intestinal failure; TLESR—transient lower esophageal sphincter relaxation; GERD—gastro-esophageal reflux disease.

**Table 3 nutrients-12-03536-t003:** Diagnostic tools in foregut dysmotility.

Investigation	Assessment	Common Findings in IF
**Videofluoroscopic swallow study (VFSS)**	Assesses anatomic abnormalities in the upper aerodigestive tract; evaluates bolus movement during swallowing and risk of aspiration	Oro-pharyngeal dysphagia Aspiration.
**Upper gastrointestinal contrast study**	Evaluates anatomic abnormalities in the upper gastrointestinal tract	Esophageal dilatation. Hiatal hernia. Malrotation. Dilated small bowel loops, with possible contrast retention.
**24 h pH-impedance**	Ambulatory, detects differentiates liquid, mixed, and gas GER events, acid and non-acid GER. Used as “Gold standard” in the GERD diagnostics.	Increased number of reflux episodes. Increased total acid exposure time.
**Upper endoscopy**	Anatomical and mucosal assessment of the upper GI tract.	Higher rate of esophageal erosive disease. Hiatal hernia Mucosal inflammation Proximal strictures
**Oropharyngeal and esophageal manometry**	Evaluates esophageal motor function and oro-pharyngeal coordination	Lower frequency and poor propagation of spontaneous swallows. Slow peristaltic propagation velocity. Aperistaltic esophageal body.
**Scintigraphy for gastric emptying**	Normal ranges available for liquid and solid meal in children	Delayed gastric emptying for liquid and solid meal. Abnormal gastric accommodation.
**Antro-duodenal manometry**	Evaluates motor function of the antrum and proximal small bowel.	Neuro-or myopathic small bowel. Absence of MMC (phase III complex). Abnormal postprandial phase. Antral hypomotility.

IF—intestinal failure; GER—gastroesophageal reflux; GERD—gastroesophageal reflux disease; GI—gastrointestinal; MMC—migrating motor complex.

**Table 4 nutrients-12-03536-t004:** Current evidence for the use of prokinetics in foregut dysmotility in children [67].

Motility Agent	Segments and Strength of Activity	Evidence
**Metoclopramide**	Esophagus + Stomach + Small bowel +	Not recommended in GERD or gastroparesis in children due to side effects (risk of extrapyramidal symptoms, tardive dyskinesia). Approved for short term treatment in adults with gastroparesis (US) [120]. Current literature is insufficient to either support or oppose the use of metoclopramide for gastroesophageal reflux disease in infants [121].
**Domperidone**	Esophagus + Stomach +	Not recommended in GERD [119,122]. 4 weeks of therapy were only minimally effective infants and children with GERD [123]. Effective in children with diabetic gastroparesis [124].
**Cisapride**	Esophagus + Stomach + Small bowel + Colon +	Withdrawn in July 2000 following cardiac adverse reactions in adults in UK. Decrease in regurgitation and acid reflux in infants [125]. Modest improvement in feeding tolerance in children with SBS and GI dysmotility [126].
**Erythromycin, Azithromycin**	Esophagus + Stomach ++ Small bowel +	No effect on feeding tolerance or GERD in preterm infants [127]. Some evidence on improving enteral feeding in preterm infants with moderate/severe GI dysmotility [128]. Erythromycin rarely induced phase III (MMC) in patients who did not have it during fasting period [129].
**Amoxicillin-clavulanate**	Small bowel +	Possible prokinetic effect through the release of intraluminal motilin or interaction of beta-lactam with postsynaptic gamma-aminobutyric acid receptors in myenteric plexus [130]. Induces phase III-type contractions in the duodenum in children [131].
**Baclofen**	Esophagus + Stomach +	Reduction of reflux episodes by reducing the number of transient lower esophageal sphincter relaxations [132,133].
**Neostigmine, Pyridostigmine**	Esophagus + Stomach + Small bowel + Colon +	Increase of GI motility by enhancing availability of acetylcholine at neuromuscular synapses, such as the myenteric plexus [134]. Case reports showing improvement in PIPO, colonic transit, esophageal motility [135,136,137].
**Octreotide**	Stomach + Small bowel + Colon +	Induction of the phase III in the small bowel, decrease in antral motility [138].
**Prucalopride**	Esophagus + Stomach + Small bowel + Colon +	Improvement of symptoms in adult patients with chronic pseudo-obstruction [139].

GERD—gastroesophageal reflux, SBS—short bowel syndrome; GI—gastrointestinal, MMC—migrating motor complex; PIPO—pediatric intestinal pseudo-obstruction syndrome, +/++—proportion of the drug receptors distribution across the gastrointestinal tract.

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
