# Peer review of "Gastroesophageal Reflux Disease and Foregut Dysmotility in Children with Intestinal Failure"

_nutrients, 2020, doi:10.3390/nu12113536_

Round 1
Reviewer 1 Report
Dear Authors,
I am glad to be a reviewer of this paper
the article entitled »Gastroesophageal Reflux Disease and Foregut Dysmotility in Children with Intestinal Failure« focused on the pathogenesis of Impaired gastrointestinal dysmotility in children.
this review is well-written and structured and represents an important contribution to clinical nutrition. Therefore, this review is important and should be published. However, I have a few concerns.
• The manuscript is in general clearly written but in some cases, there were errors that should be checked i.e:
Line 116 esophageal dysmotility. [29]. Dot before brackets should be deleted
Please see: https://www.mdpi.com/journal/nutrients/instructions
Line 135 with a reported incidence for the general 0-5 years pediatric population (5.5%)[34] missed space between brackets
• Please add information about the influence of genetics on the development of the disease
• Conclusion section is too short and should be extended.
• Review is valuable and contains novelty, but only ca. 13 % of the references are from 2018-2020. (ca. 20 publication from a total of 149) Please clarify why?
• Methodology: How did you search the articles for your work, what kind of keywords and database did you use?
Author Response
Please, see the attachment.

Reviewer 2 Report
This narrative review provides a detailed overview of the mechanisms contributing to GER and foregut dysmotility in children with intestinal failure (IF) and short bowel syndrome (SBS) including information on practical applications of this knowledge to aid diagnostic and management decisions for clinicians working in this specialist area.
It is a useful topic with important potential clinical applications and limited previous publications focussed on this area. Overall, the content is detailed and includes a large number of references to previous related literature. There are some minor grammatical and content changes required that will improve the flow and ease of readability of the article (detailed below).
The use of abbreviations needs to be reviewed in multiple lines, as this is sporadic and inconsistent throughout the article, especially for intestinal failure (IF), short bowel syndrome (SBS), parenteral nutrition (PN), necrotising enterocolitis (NEC) and gastroesophageal reflux disease (GERD). These should be defined in the introduction at first use and changed to abbreviation from then on, apart from tables.
Table 1: suggest review title for column 2. This column lists the mechanisms leading to impact on GI motility rather then the impact on GI motility itself. Consider rename to 'Possible mechanisms contributing to GI dysmotility'
Line 66: access to investigations in this area is very dependent on local resources / expertise. Understanding of pathophysiology helps to guide clinicians and prioritise appropriate selection of investigations (however depends on consideration of resources available to unit), but also avoids unnecessary use of investigations / management options.
Line 89: suggest insert 'for example' after oral aversive stimuli
Line 98: insert 'an' before important
Line 99: remove 'also' before social factors
Line 103: insert 'and' before contribute
Line 106 to 110: it is not clear whether this study of 700 children was specifically children with IF or a general paediatric population. This makes this paragraph and its purpose a bit confusing.
Line 122: GER has already been discussed. The definition given here may be better suited earlier in the article when GER / GERD first mentioned.
Table 2: Suggest to align Table title with header for column 3- SBS is a subset of IF but it is not clear why SBS has been specifically included as opposed to non-anatomical causes of IF and may create some confusion. It may be better to keep table more generally focussed on IF rather than include only some specific subtypes or otherwise to break up into SBS vs non-SBS.
Line 130 to 133: Widespread use of PPIs and H2R antagonists is now common practice from an early age (post resection) which may be an influence on the limited availability of data on pH probes in this population as well as epidemiology on erosive esophagitis if already treated.
Line 165: abbreviation for intestinal malrotation introduced although not used consistently before or after this.
Section II: Regurgitation and GER in children may benefit from use of further sub-headings to provide more structure to text.
Line 228: similar to section II / GER, this definition of SBS would be better placed earlier in the article (e.g. start of section I) to define the condition earlier as it has been mentioned multiple times.
Line 238 to 239: 'Antro-duodenal and colonic manometry in SBS are not being performed regularly'. This statement needs to be substantiated by reference and / or explanation of clinical relevance e.g. whether this is a good or bad thing, is this resource driven or because of limitations to administering / interpreting these tests...?
Line 274 and 293: extend should be extent
Line 281: remove comma after both
Line 321: Grammar: change 'when' to 'while' or similar.
Line 355: Grammar: change 'dependent to' to 'dependent on'
Table 3: Column 3 title suggest add 'common' or 'possible' findings in IF / SBS
Again, similar to Table 2, it is not clear why SBS is included specifically in this table and not non-anatomical IF. Most likely, this is related to the fact that there is a lot more literature available on SBS rather than non-SBS IF however, given the article title / aim is focussed on IF then the Tables should ideally align with this or otherwise include reference to both non-SBS as well as SBS causes of IF.
Line 390: reference to other nutritional management options described in paragraph above- it is not clear whether this is referring to strategies for minimising oral aversion or something else? This section (I Non-pharmacological and dietary treatment) is rather brief, especially given pharmacological options have limited outcomes / application so appropriate application of non-pharmacological and dietary treatment becomes essential. Could benefit from additional detail and / or reference to appropriate guidelines e.g. NASPGHAN / ESPGHAN GERD mentioned in next section.
Line 403: use abbreviation PPIs
Section II:
Would be good to include some discussion of recommended duration of prokinetic trials of in clinical practice and how response is monitored.
Author Response
Please, see the attachment.

Reviewer 3 Report
- “Treatment of gastric acid hypersecretion with H2 receptor antagonists and/or proton pump inhibitors has improved nutrient absorption with short bowel syndrome.”
Why?
- “liquid nutrient component of meals …. should be advocated in patients with delayed gastric emptying”
Does not liquid component in stomach worse reflux?
- If PPI are superior to anti-H2, why use anti-H2?
- What about buspirone or amitriptyline (see Caviglia GP et al. Gastric emptying and related symptoms in patients treated with buspirone, amitriptyline or clebopride: a "real world" study by 13C-octanoic Acid Breath Test. Minerva Med. 2017 Dec;108(6):489-495. doi: 10.23736/S0026-4806.17.05320-4. Epub 2017 Jul 12. PMID: 28707862.)?
- What about chronic constipations in these patients?
Author Response
Please, see the attachment.
